# Curcuminoids Inhibit Angiogenic Behaviors of Human Umbilical Vein Endothelial Cells via Endoglin/Smad1 Signaling

**DOI:** 10.3390/ijms23073889

**Published:** 2022-03-31

**Authors:** Yi-Fan Chou, Yu-Hsuan Lan, Jun-Han Hsiao, Chiao-Yun Chen, Pei-Yu Chou, Ming-Jyh Sheu

**Affiliations:** 1Graduate Institute of Biomedical Sciences, College of Medicine, China Medical University, Taichung 411710, Taiwan; yifanchou29@hotmail.com; 2Department of Otolaryngology, Buddhist Tzu Chi Medical Foundation, Taichung Tzu Chi Hospital, Taichung 411710, Taiwan; 3School of Medicine, Tzu Chi University, Hualien 970020, Taiwan; 4School of Pharmacy, College of Pharmacy, China Medical University, Taichung 406040, Taiwan; febmov1811@gmail.com (J.-H.H.); restdosidaemita@gmail.com (C.-Y.C.); 5Bachelor of Science in Senior Wellness and Sports Science, Tunghai University, Taichung 411710, Taiwan; peiyu67@gmail.com; 6College of Active Aging and Technology, Tunghai University, Taichung 411710, Taiwan

**Keywords:** angiogenesis, curcuminoids, endoglin, Smad1, VEGF

## Abstract

Background: Angiogenesis is primarily attributed to the excessive proliferation and migration of endothelial cells. Targeting the vascular endothelial growth factor (VEGF) is therefore significant in anti-angiogenic therapy. Although these treatments have not reached clinical expectations, the upregulation of alternative angiogenic pathways (endoglin/Smad1) may play a critical role in drug (VEGF-neutralizing agents) resistance. Enhanced endoglin expression following a VEGF-neutralizing therapy (semaxanib^®^) was noted in patients. Treatment with an endoglin-targeting antibody augmented VEGF expression in human umbilical vein endothelial cells (HUVECs). Therefore, approaches that inhibit both the androgen and VEGF pathways enhance the HUVECs cytotoxicity and reverse semaxanib resistance. The purpose of this study was to find natural-occurring compounds that inhibited the endoglin-targeting pathway. Methods: Curcuminoids targeting endoglin were recognized from two thousand compounds in the Traditional Chinese Medicine Database@Taiwan (TCM Database@Taiwan) using Discovery Studio 4.5. Results: Our results, obtained using cytotoxicity, migration/invasion, and flow cytometry assays, showed that curcumin (Cur) and demethoxycurcumin (DMC) reduced angiogenesis. In addition, Cur and DMC downregulated endoglin/pSmad1 phosphorylation. Conclusions: The study first showed that Cur and DMC demonstrated antiangiogenic activity via the inhibition of endoglin/Smad1 signaling. Synergistic effects of curcuminoids (i.e., curcumin and DMC) and semaxanib on HUVECs were found. This might be attributed to endoglin/pSmad1 downregulation in HUVECs. Combination treatment with curcuminoids and a semaxanib is therefore expected to reverse semaxanib resistance.

## 1. Introduction

Tumor angiogenesis is a key process mainly found in tumors that develop to be larger than 1–2 mm in diameter, which will generally stimulate tumor metastasis. Thus, tumor angiogenesis has been suggested as a useful target whose appearance can be used as a tool to fight all solid cancers [1]. Targeting the vascular endothelial growth factor (VEGF) is the principal therapy for angiogenesis. However, these treatments do not show satisfactory clinical outcomes; the reason may be the upregulation of endoglin/Smad1 alternative angiogenic signaling. Evidence shows that enhanced endoglin (CD105) expression following a VEGF-neutralizing therapy (semaxanib^®^) was noticed in patients [2]. Semaxanib^®^ (SU5416) is a small molecule exhibiting potent and selective inhibition of VEGF receptor-2 tyrosine kinase [3]. In addition, one study suggests that endoglin-neutralizing antibody significantly augments VEGF expression in HUVECs [2]. The upregulation of TGFβ1/endoglin, an angiogenesis promoter, was found after treatment with bevacizumab. Elevated endoglin caused the activation of Smad1/5, which plays a role in the inflammation path and the endothelial–mesenchymal transition. The migration capability of HUVECs was augmented by bevacizumab treatment [4].

Endoglin, a co-receptor for tumor growth factor-β (TGF-β) in HUVECs, plays an important role in tumor angiogenesis [5]. Endoglin controls angiogenesis through the regulation of phosphorylation Smad1, causing cell migration and adhesion by changing the structure of focal adhesion complexes. Abnormal functions of endoglin are important for various cellular processes implicated in cancer [6]. In addition, tumor-related angiogenesis may be caused by endoglin dysregulation [7]. The overexpression of endoglin is detected in angiogenesis, progression, and metastasis of tumors [8]. The inhibition of endoglin may be beneficial for cancer treatment [9]. The TRC105 antibody (Carotuximab^®^) demonstrates a high affinity for endoglin and may overcome therapeutic resistance when combined with bevacizumab [10]. Recently, many clinical trials have supported the possibility that endoglin-targeting potentiates anti-VEGF therapies [11,12,13].

Curcuminoids, derived from *Curcuma longa* Linn, comprise curcumin (Cur; 77%), demethoxycurcumin (DMC; 17%), and bisdemethoxycurcumin (bDMC; 3%) [14]. Curcumin has been suggested to suppress tumor initiation, promotion, metastasis [15,16,17], and VEGF inhibition [18]. Curcumin was shown to reduce human lung cancer migration and invasion by MMP-2/MMP-9 inhibition and VEGF suppression [19]. Our research has shown that co-administration with curcuminoids boosts cisplatin’s cytotoxicity in lung cancer [20,21]. DMC exhibited the strongest efficacy to inhibit vascular smooth muscle cell migration and neointima formation induced by a balloon injury [22]. Additionally, curcuminoids showed the inhibition of p-glycoprotein and the reversal of multidrug resistance [23]. Unlike curcumin, little of which is absorbed after oral consumption, DMC demonstrated better stability and aqueous solubility at physiological pH [24]. A previous study indicated that DMC significantly repressed the capillary network formation in the aorta of rats [25] and inhibited ex vivo neovascularization of chick chorioallantoic membrane (CAM). Other studies have revealed that MMP-9 participated in inhibiting angiogenesis by DMC. MMP-9 has been shown to play a key role in the progress of angiogenesis [26,27]. Studies have verified that DMC markedly represses the migration and invasion of HUVECs. However, it remains elusive whether curcuminoids have the capacity to inhibit the endoglin overexpression after the VEGF-neutralizing agent.

Accordingly, the inhibition of both endoglin and VEGF signaling increases HUVEC cytotoxicity and may reverse semaxanib resistance. The purpose of this study was to identify which of the naturally-occurring compounds inhibit the endoglin-targeting pathway. We expect that these natural compounds could be further used together with VEGF-neutralizing agents to treat cancer patients.

## 2. Results

### 2.1. Binding to Endoglin by Curcuminoids Was Recognized by Discovery Studio 4.5 (D.S. 4.5)

D.S. 4.5 was applied to the virtual screening of many compounds to target the endoglin. Structures of all two thousand of these compounds from the TCM Database @Taiwan were employed for the screening process. Among these eight hundred more active endoglin-targeting compounds, curcuminoids (i.e., curcumin, DMC, and bDMC) exhibited good scores for -CDOCKER ENERGY and -CDOCKER INTERACTION ENERGY (Table 1). The -CDOCKER energy includes all the interaction energy between the curcuminoids (i.e., Curcumin, DMC, and bDMC), the corresponding receptor, and their internal ligands. However, the -CDOCKER interaction energy explains the interaction energy between the ligand and the endoglin protein in simple terms. Curcuminoids (i.e., curcumin, DMC, and bDMC) were assessed for their binding affinity with endoglin, and their values of GI_50_ are recorded in Table 1. Using D.S. 4.5, curcumin, DMC, and bDMC were confirmed to comparatively bind to endoglin (Figure 1A–C and Table 1). In line with the docking analysis, curcumin was shown to exhibit higher binding affinity to endoglin as regulated by hydrogen bonds with ARG121, TRP117, ARG399, LYS97, LYS70, and Glu100 and Pi interacting with GLU99 (Figure 1A). Additionally, DMC showed higher binding affinity to endoglin as mediated by specific hydrogen bonds with GLU100, TYR210, ASN67, LYS7, and GLU166 and Pi interacting with ARG121 and ARG399 (Figure 1B). These bonds are critical for the interaction between the endoglin and curcumin and DMC, respectively. Our results showed that the -CDOCKER energies of curcumin, DMC, and bDMC were −55.3049 kcal/mol, −55.2537 kcal/mol, and 48.5405 kcal/mol, respectively. These data show that curcuminoids (i.e., curcumin, DMC, and bDMC) bind to the endoglin.

### 2.2. Inhibitory Effect of Curcuminoids (Curcumin, DMC, and bDMC) Alone and in Combination Treatment with Semaxanib (SU5416) on Cytotoxicity of HUVECs

The cytotoxic effect of curcuminoids on HUVECs was examined. Our results demonstrate that curcumin (Figure 2A) and DMC (Figure 2B) significantly increased the HUVEC cytotoxicity after 72 h of treatment. In this study, curcuminoids significantly inhibited the cell viability in a dose-dependent manner, and the GI_50_ values of curcumin, DMC, and bDMC on HUVECs were about 11.11, 18.03, and >100 μM, respectively (Figure 2A–C). Other than bDMC, lower concentrations of curcumin (0.625–1.25 µM) and DMC (0.625–1.25 µM) were then applied to determine their biological activity. Our results suggest that curcumin (0.625–1.25 µM) synergistically increased the anti-angiogenesis of semaxanib (25 µM) in endothelial cells (Figure 2E). In addition, DMC (0.625–1.25 µM) improved the anti-angiogenesis of semaxanib (25 µM) in endothelial cells (Figure 2F, Table 2). The coefficient of drug interaction (CDI) was calculated with the equation CDI = AB/(A × B) (AB, relative cell viability treated with the combination (i.e., curcumin/DMC and semaxanib); A or B, the relative cell viability of the groups treated with the single agents). CDI < 1 reveals a synergistic effect, CDI = 1 reveals an additive effect, and CDI > 1 reveals an antagonistic effect. (Table 2 and Table 3).

### 2.3. Inhibitory Effect of Curcuminoids (Cur, DMC, and bDMC) on the Cell-Cycle Distribution of Endothelial Cells

The arrest of the cell cycle could regulate cell growth. To examine the inhibitory effect of curcuminoids on the growth of endothelial cells, flow cytometry was used for the study. HUVECs were administered with curcumin (0.625, 1.25, 2.5, 5.0, 10.0, and 20.0 μM), and DMC (0.625, 1.25, 2.5, 5.0, 10.0, and 20.0 μM) for 72 h, and then their DNA content was examined for 10,000 events. The rise of the sub-G1 peak and the buildup of endothelial cells in the G0/G1 phase showed a dose-dependent decline in the S phase (Figure 3A,B).

### 2.4. Curcuminoids Reduced the In Vitro Angiogenic Activity of Endothelial Cells

#### 2.4.1. Curcumin Hinders the Migration and Invasion of Endothelial Cells

The activities of endothelial migration and invasion represent the early stage in the development of neo-peritumoral blood vessels through angiogenesis. However, higher doses (5–20 µM) of curcumin significantly decreased the migration and invasion of HUVECs. Our results show that low doses of curcumin (0.625–2.5 µM) significantly diminished the migration of HUVECs treated with 10% FBS (Figure 4A). In addition, the results showed that low-dose curcumin (0.625–2.5 µM) meaningfully reduced the invasion of HUVECs treated with 10% FBS (Figure 4B).

#### 2.4.2. DMC Reduces the Migration and Invasion of Endothelial Cells

The migration and invasion of endothelial cells are necessary for angiogenesis and are critical in the early stage of the peritumoral blood vessel formation during angiogenesis. However, higher doses (5–20 µM) of DMC substantially decreased the migration and invasion of HUVECs. The current study confirmed that low doses of DMC (0.625–2.5 µM) considerably decreased the migration of HUVECs treated with 10% FBS (Figure 4C). In addition, the results showed that low doses of DMC (0.625–2.5 µM) considerably reduced the invasion of HUVECs treated with 10% FBS (Figure 4D).

### 2.5. Curcuminoids Decreased Endoglin and Smad1 Phosphorylation in HUVECs

The effects of curcumin and DMC on endoglin and phosphorylation Smad1 protein expressions were explored in HUVECs. Endoglin is believed to be involved in the formation of neo-vessels, and VEGF is critical to initiate the vessel formation. Consequently, our research aimed to find out if low doses of curcumin (0.625–5.0 µM) and DMC (0.625–5.0 µM) could modulate the endoglin and phosphorylation Smad1 levels in HUVECs. Our results demonstrate that long-term exposure (72 h) to curcumin and DMC considerably decreased the endoglin levels and phosphorylation of Smad1 in HUVECs. In particular, the endoglin/pSmad1 protein expression was normalized by GAPDH and found to be significantly decreased (Figure 5A,B).

## 3. Discussion

The vascular development of tumors is necessary for tumor metastasis and is, therefore, a major target of anti-angiogenic therapy [1,28]. Therefore, targeting VEGF is used for anti-angiogenic therapy. However, VEGF-neutralizing treatment cannot attain clinical outcomes, and the upregulation of endoglin/Smad1 signaling may play an important role in VEGF-neutralizing agents’ resistance. Increased endoglin expression following a VEGF-neutralizing therapy (semaxanib^®^) was found in vivo. Additionally, an endoglin-targeting antibody enhanced VEGF expression in HUVECs [2]. Thus, combination treatment with an endoglin antagonist and with VEGF-neutralizing agents may strengthen anti-angiogenic therapy.

Although endoglin plays an important role in regulating angiogenesis, the present study demonstrated that overexpression of endoglin does not improve angiogenesis. Nevertheless, it prevents blood vessels from maturing and stabilizing during angiogenesis. Contrary to what was postulated, endoglin overexpression does not cause an increase in tumor vascularization that expedites the intravasation and metastasis of tumor cells [29]. However, most studies believe that endoglin is overexpressed in angiogenic endothelial cells. Research has indicated that endoglin knockout mice could develop embryonic lethality in an animal study. Therefore, endoglin is essential for angiogenesis in the development phase [30]. Previous preclinical research and clinical trials indicate that endoglin is an important biomarker of angiogenesis. Additionally, the overexpression of endoglin has been detected in the angiogenic vasculature [31], which unambiguously shows that endoglin serves as a positive regulator of the angiogenic pathway in HUVECs [32]. Before phosphorylation of Smad1, endoglin activates its intracellular signaling cascade, transmits the signals into the nucleus, and transcribes a variety of genes [33]. According to the data from a genetic analysis, BMP-9 is essential for the development of vessels in HUVECs. Research indicates that TGF-β promotes both ALK1 and ALK5 type I receptors in endothelial cells; nevertheless, BMP-9 only binds to ALK1. This is because BMP-9 shows a higher affinity for ALK1 than for TGF-β. Smads 1/5/8 and Smads 2/3 are phosphorylated specifically by ALK1 and ALK5. Subsequently, Smads 1/5/8 are translocated into the nucleus and later modulate the expression of genes related to cell motility, proliferation, adhesion, apoptosis, and angiogenesis [34,35]. Consequently, endoglin/Smad1 signaling influences the angiogenesis of HUVECs. Clinically, many ALK1 inhibitors are used for renal and ovarian carcinomas. Together with VEGF signaling obstruction, ALK1 inhibitors meaningfully prevent tumor progression by angiogenesis [36]. Endoglin has been found to be overexpressed in tumor-associated angiogenic vessels compared with normal vessels [37,38]. TRC105, an IgG1 endoglin monoclonal antibody (MAb), has been found to diminish tumor metastasis by inhibiting endoglin-modulated angiogenesis [4].

VEGF signaling is crucial for tumor angiogenesis. Clinically, bevacizumab exhibits limited impacts caused by the development of acquired resistance [2]. According to a previous report, TRC105 (Carotuximab^®^)-treated patients demonstrated major downregulation of VEGF. Overexpression of endoglin was also noticed in patients following VEGF-neutralizing treatment [37]. Therefore, endoglin/Smad1 signaling may be correlated with the development of drug resistance [2,38]. Combination treatment with an endoglin antagonist and VEGF antagonist enhances the inhibition of HUVEC angiogenesis, which overcomes drug resistance to VEGF-neutralizing agents. Combination therapy with TRC105 and decitabine provides a sustained anti-leukemic impact in acute myeloid leukemia (AML) xenografts compared to decitabine alone [39]. TRC105 enhanced the inhibitory effect of sunitinib on VEGF-VEGFR2-Akt-Creb signaling, indicating a molecular collaboration between TRC105 and Sunitinib [40]. Although TRC105 failed to improve progression-free survival (PFS) when added to bevacizumab [12,41], other studies showed promising clinical outcomes. In several recent clinical studies, combination therapy with TRC105 and VEGF-neutralizing agents has been demonstrated. Together with TRC105, bevacizumab showed significant clinical outcomes in a VEGF-inhibitor-refractory population [42]. When administered TRC105 and sorafenib, patients showed good toleration of both drugs at the suggested single-agent doses in a phase I and a preliminary phase II study [43]. In the phase Ib trial, patients treated with both TRC105 and bevacizumab showed improved clinical outcomes [44]. In the combination therapy with TRC105 and pazopanib, patients exhibited stable complete responses and inspired progression-free survival [11]. A randomized Phase II trial test showed that VEGF-inhibitor-refractory renal-cell-carcinoma patients given TRC105 and axitinib together showed a stable condition [13].

Although protein drugs have shown promising outcomes for antagonizing tumor angiogenesis, there are several drawbacks, including a higher cost, stability, and severe immune-related adverse effects. Therefore, small molecules used to modulate the endoglin/Smad1 signaling must be established, along with combination treatment with a VEGF antagonist to decrease drug resistance. Our results demonstrate that combination therapy with curcuminoids (i.e., curcumin and DMC) with semaxanib creates anti-angiogenic effects (Table 2 and Table 3). These results were verified by previous research. One study suggested that toxicarioside A could hinder tumor development via endoglin [45,46]. Our study indicates that the combined use of EGCG and semaxanib could solve the drug resistance to VEGF-neutralizing agents [27,47]. Nevertheless, the pharmacological effect of angiogenic activity by curcuminoids has not yet been studied. In this study, we aimed to explore naturally occurring compounds that could inhibit this alternative proangiogenic pathway. These results demonstrate that lower-dose curcumin and DMC may inhibit endothelial cell viability (Figure 2A,B). In addition, curcumin- and DMC-treated HUVECs showed an increase in Sub-G1 peak, the accumulation of endothelial cells in the G0/G1 phase, and the reduction in the S phase (Figure 3A,B). Our results show that lower doses of curcuminoids (i.e., curcumin and DMC) substantially inhibit the migration/invasion of endothelial cells (Figure 4). Furthermore, curcuminoids synergistically enhance the angiogenesis of semaxanib in endothelial cells (Table 2 and Table 3). In particular, our results showed that both curcumin and DMA significantly reduce endoglin/pSmad1 protein expression (Figure 5A,B). According to a previous study, the redundancy between endoglin and VEGF pathways in angiogenesis and the effects of targeting both pathways [2], as well as endoglin-targeted inhibition, could enhance the angiogenic activity of VEGF-neutralizing agents. This research strengthens our understanding that the combination of endoglin-antagonists and VEGF-neutralizing agents could enhance the anti-angiogenic effects of therapy.

## 4. Materials and Methods

### 4.1. Chemicals and Reagents

The major cell culture media, Medium 199 (M199), phosphate-buffered saline (PBS), trypsin-EDTA, and fetal bovine serum (FBS), were acquired from GIBCO (Gaithersburg, MD, USA). Semaxanib^®^ was obtained from Surgen Inc. (Redwood City, CA, USA). Tris base, heparin, trichloroacetic acid (TCA), and a bovine serum albumin (BSA) protein assay kit were purchased from Sigma-Aldrich (St. Louis, MO, USA). Curcumin, demethoxycurcumin, and bisdemethoxycurcumin were acquired from Sigma-Aldrich (St. Louis, MO, USA). Penicillin and streptomycin were acquired from Lonza (P/S; Walkersville, MD, USA). Endothelial cell growth supplements (ECGS) and polyvinylidene fluoride (PVDF) membranes were obtained from Millipore (Billerica, MA, USA). Matrigel^®^ basement membrane matrix (#356237) was acquired from Corning (Bedford, MA, USA). The primary anti-endoglin antibody was obtained from BioLegend (San Diego, CA, USA), the mouse monoclonal GAPDH antibody was purchased from Proteintech (Rosemont, IL, USA). The phospho-SMAD1/SMAD5 (pSer463 + pSer465) antibody was acquired from Thermo Scientific (Rochester, NY, USA). The goat anti-mouse IgG and goat anti-rabbit IgG secondary antibodies were obtained from Jackson ImmunoResearch Laboratories (West Grove, PA, USA). Semaxanib was purchased from Medchem Express (Monmouth Junction, NJ, USA). ThinCerts™ Cell Culture Inserts with an 8 μM pore size, #662638 were obtained from Greiner Bio-One Inc., (Monroe, NC, USA). Curcumin and DMC were solubilized in DMSO and had stock concentrations of 100 mM.

### 4.2. Cell Culture

The human umbilical vein endothelial cells (HUVEC; BCRC no. H-UV001) were obtained from the Bioresource Collection and Research Center (BCRC; Hsinchu, Taiwan). Firstly, the gelatin-coated culture dishes were prepared, the HUVECs were seeded on those dishes and grown in medium 199 containing 25 U/mL heparin (Louis, MO, USA), 100 U/mL penicillin/streptomycin (P/S; Walkersville, MD, USA), 10% FBS, and 30 mg/L ECGS. Only passages 3–5 were allowed for the experiments. The cells were incubated in a humidified 5% CO_2_ atmosphere at 37 °C. The culture medium was changed every two to three days.

### 4.3. Sulforhodamine B (SRB) Assay

The SRB assay was performed to measure the cytotoxicity of curcuminoids on HUVECs following our former study [27]. Firstly, gelatin-coated 96-well plates were used to grow the HUVECs for 30 min. Next, HUVECs were incubated with several different concentrations of curcumin alone (3.125, 6.25, 12.5, 25, 50, and 100 μM), DMC (3.125, 6.25, 12.5, 25, 50, and 100 μM), or BDMC (3.125, 6.25, 12.5, 25, 50, and 100 μM) or combined with semaxanib (25 μM), and then they were incubated in a humidified 5% CO_2_ atmosphere at 37 °C for 72 h. Only curcumin and DMC were used to treat with semaxanib (25 µM) in this research setting. Twenty-five microliters (50% TCA) were added to each well, incubated for 30 min, and then washed twice with deionized distilled water (ddH_2_O). Next, the plates were air-dried and stained with 50 μL (0.04% SRB) for 30 min. Next, they were washed twice with 1% acetic acid to remove the unbound stain. Finally, the absorbance of each well was measured at 515 nm using an ELISA reader.

### 4.4. Western Blotting Analysis

The experiments were performed as previously described [27]. HUVECs were seeded on a 6 cm dish and incubated with various dosages (0.625, 1.25, 2.5, 5, and 10 µM) of curcumin and DMC for 72 h. A BSA protein assay kit was used to determine the protein concentration. Then, cell lysates including 20 μg of protein were separated on 10% SDS-polyacrylamide gels (SDS-PAGE) and electrophoretically transferred to PVDF membranes. The membranes were incubated with the human endoglin antibody (1:1000 dilution), mouse monoclonal GAPDH antibody (1:5000 dilution), or phosphor-smad1/smad5 (pSer463 + pSer465) antibody (1:250 dilution) at 4 °C overnight. Then, the membranes were incubated with horseradish peroxidase-conjugated goat anti-mouse IgG (1:5000 dilution) or goat anti-rabbit IgG (1:5000 dilution) secondary antibodies for 1 h. Lastly, membranes were washed with PBST buffer and visualized using Amersham ECL Advance Western Blotting Detection Reagents (Millipore, Bedford, MA, USA). The luminescence signal was acquired using an LAS-4000 system (Fujifilm, Valhalla, NY, USA), and protein levels were detected using an enzymatic chemiluminescence kit.

### 4.5. Cell Migration and Invasion Assays

The experiments were performed following our previous study [27]. To evaluate the migration of HUVECs, HUVECs (2 × 10^5^ cells/well) were seeded in the upper chamber and placed in the lower chamber. To study the invasion, HUVECs (3 × 10^5^ cells/well) were pre-mixed with the same volume of thawed Matrigel and immediately seeded in the upper chamber. In the experimental setting, curcumin (0.625, 1.25, 2.5, 5, and 10 µM) or DMC (0.625, 1.25, 2.5, 5, and 10 µM) was added to the upper chambers, and 10% FBS at the lower chamber could stimulate migration and invasion of endothelial cells. After six hours, the culture medium (or Matrigel) was scratched from the upper chamber, and 99% methanol was used to fix the migrated or invaded cells attached to the outer surface of the upper chamber for twenty min at room temperature. Then, these fixed cells were stained with Giemsa stain. We randomly chose five regions to count the cell number using a light microscope at 100× magnification. The cell number of the control group (treated with 10% FBS alone) was set to 100%.

### 4.6. Cell Cycle Analysis

The research was conducted following our previous study [27]. Briefly, various concentrations of curcumin (0.625, 1.25, 2.5, 5.0, 10.0, and 12.0 μM) and DMC (0.625, 1.25, 2.5, 5.0, 10.0, and 12.0 μM) were given to the HUVECs (5 × 10^5^) for 72 h in the treated and normal samples. Then, cells were washed twice with PBS, and staining was conducted by incubating the HUVECs with 500 μL of 4 mg/mL DAPI for 15 min in the darkroom. Then, the distribution of the cell cycle was recognized by flow cytometry using the FACSCanto system (BD Biosciences, San Jose, CA, USA). Their DNA content of 10,000 events was examined. Data analysis was performed using ModFit LT software (Verify Software House, Topsham, ME, USA). All the experiments were performed three times.

### 4.7. Molecular Docking Studies Were Conducted to Investigate the Binding Mode of Curcuminoids

The perfect crystal structure of the endoglin from humans (PDB ID: 5HZW_A), obtained from the NCBI databases (https://www.ncbi.nlm.nih.gov/protein/ (accessed on 30 May 2017), was applied for the molecular docking analysis employed utilizing the Discovery Studio (version 4.5). A setting of more than two thousand compounds from the TCM Database@Taiwan, which includes 61,000 compounds, was utilized to make ligands and create 3D conformations. All hydrogen atoms were added to the protein using D.S. 4.5. The CHARMm was designated, and the active site was described. Next, using the CDOCKER protocol, the optimized ligands were docked with the endoglin protein. Lastly, compounds curcumin, DMC, and bDMC gained from the virtual screen then carried out their angiogenic activity in vitro.

### 4.8. Statistical Analysis

The statistical analysis was performed using the SPSS Statistics software (vers. 19; IBM, Armonk, NY, USA). All the independent values are represented as means ± standard deviations (SD). In the multiple-group tests, a one-way analysis of variance (ANOVA) followed by a post hoc test (Dunnett’s test) was run to verify the statistical significance. A *p*-value < 0.05 was believed to be statistically significant, * and ^#^ for *p* < 0.05, ** and ^##^ for *p* < 0.01, and *** and ^###^ for *p* < 0.001.

## 5. Conclusions

Endoglin overexpression in HUVECs following semaxanib treatment was found to explain drug resistance. The results showed that curcumin and DMC may target endoglin and then show the ability to inhibit HUVECs angiogenesis via endoglin/pSmad1 signaling. Above all, the regulation of VEGF and endoglin pathways enhances the anti-angiogenic capacity in vitro. In line with the redundancy between endoglin and VEGF signaling in angiogenesis, the novelty of this study is that treatment with curcuminoids significantly decreases endoglin/Smad1 signaling. These results suggest the possibility of combination treatment with curcuminoids and semaxanib. Additionally, the combination treatment with endoglin and VEGF would be applicable for conquering resistance to VEGF-neutralizing agents.

## Figures and Tables

**Figure 1 ijms-23-03889-f001:**
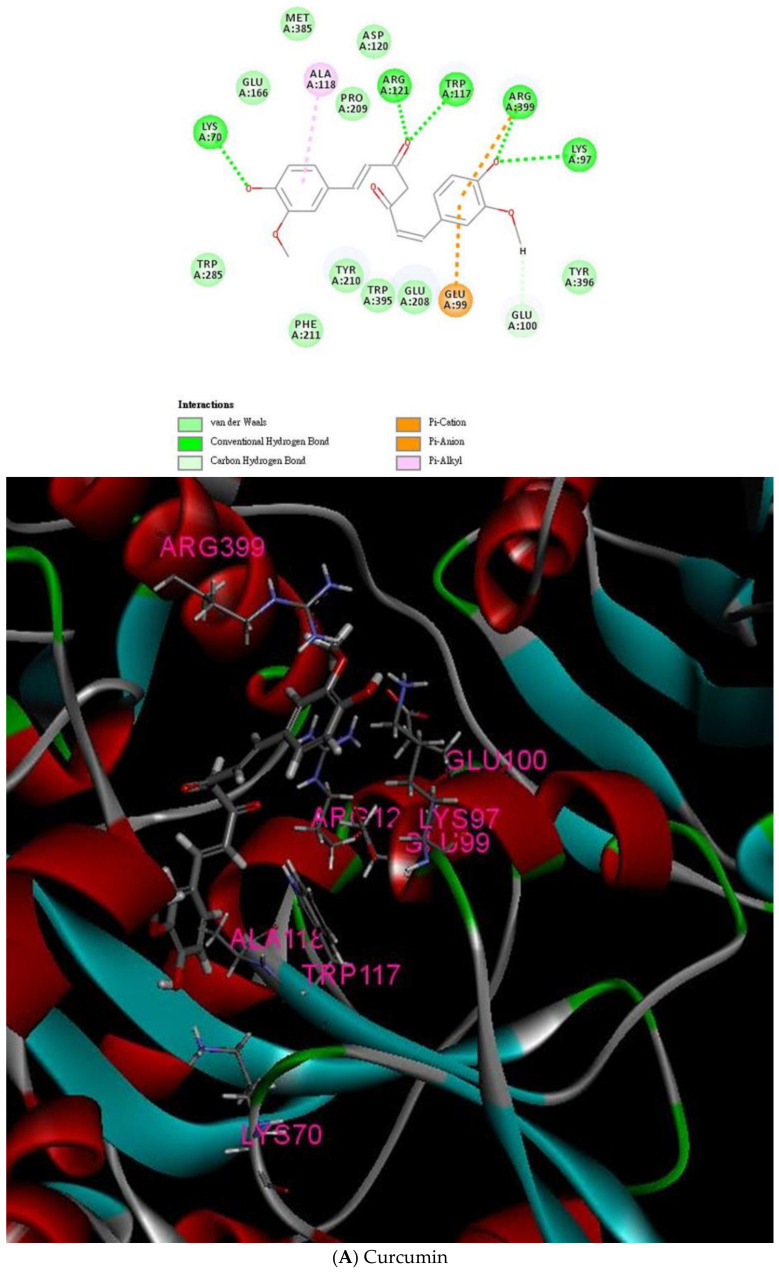
Binding mode of the interactions between curcuminoids (i.e., curcumin, DMC, and bDMC) and the endoglin residues. (**A**) Left panel: interaction between curcumin and endoglin (2D visualization). Right panel: interaction between curcumin and endoglin adduct (3D visualization). (**B**) Left panel: interaction between DMC and endoglin (2D visualization). Right panel: interaction between DMC and endoglin adduct (3D visualization). (**C**) Left panel: interaction between bDMC and endoglin (2D visualization). Right panel: interaction between bDMC and endoglin adduct (3D visualization).

**Figure 2 ijms-23-03889-f002:**
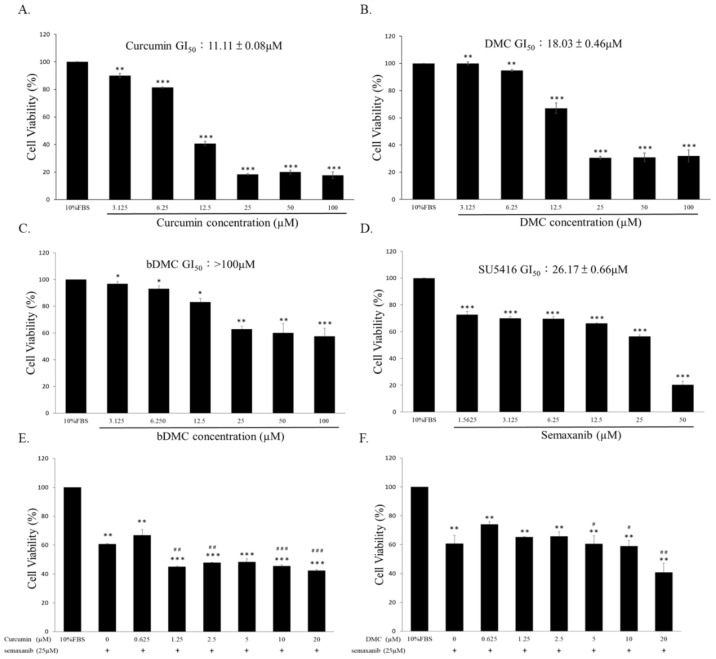
Inhibitory effects of curcuminoids (i.e., curcumin, DMC, and bDMC) alone and combined with semaxanib on HUVECs. (**A**–**C**) Cell-cytotoxic effects of curcumin, DMC, and bDMC on HUVECs according to SRB assay. (**D**) Cell-cytotoxic effects of semaxanib on HUVECs. (**E**) Combination treatment with curcumin and semaxanib demonstrated significant inhibitory effects on HUVECs compared with the semaxanib-treated group. (**F**) Combination treatment with DMC and semaxanib showed considerable inhibitory effects on HUVECs compared to the semaxanib-treated group. * *p* < 0.05, ** *p* < 0.01, and *** *p* < 0.001 compared to the control group (treated with 10% FBS alone). ^#^
*p* < 0.05, ^##^
*p* < 0.01, and ^###^
*p* < 0.001 compared to the semaxanib-treated group.

**Figure 3 ijms-23-03889-f003:**
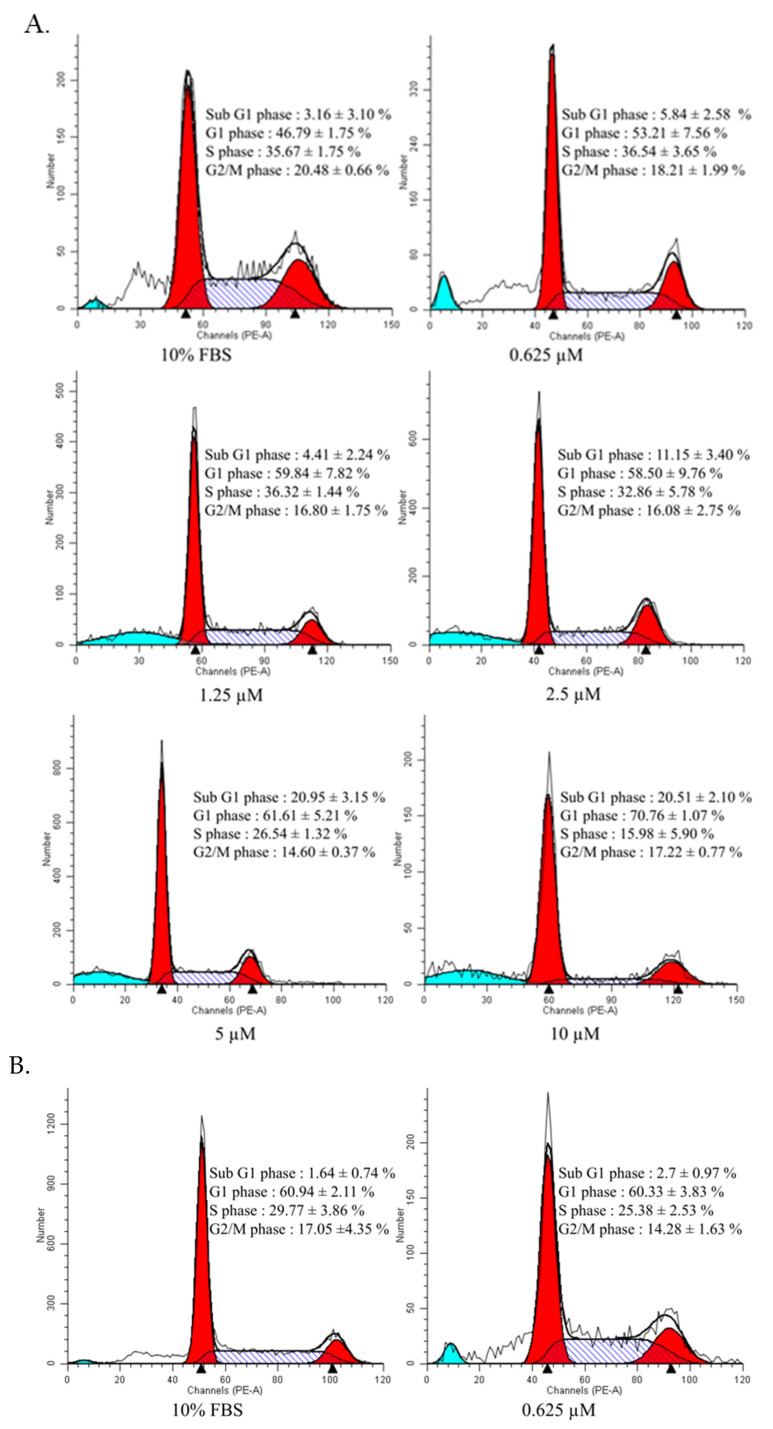
Effect of curcuminoids (i.e., curcumin and DMC) on the cell cycle distribution of HUVECs. (**A**) The cell-cycle distributions were analyzed in the HUVECs treated with curcumin. The cells were treated with several concentrations (0, 0.625, 1.25, 2.5, 5, and 10 µM) of curcumin for 72 h, and flow cytometry was used to analyze cell cycle distributions. (**B**) The cell cycle distributions were assessed in the HUVECs administered with DMC. The cells were administered with several different concentrations (0, 0.625, 1.25, 2.5, 5, 10, and 20 µM) of DMC for 72 h, and cell cycle distributions were analyzed by flow cytometry.

**Figure 4 ijms-23-03889-f004:**
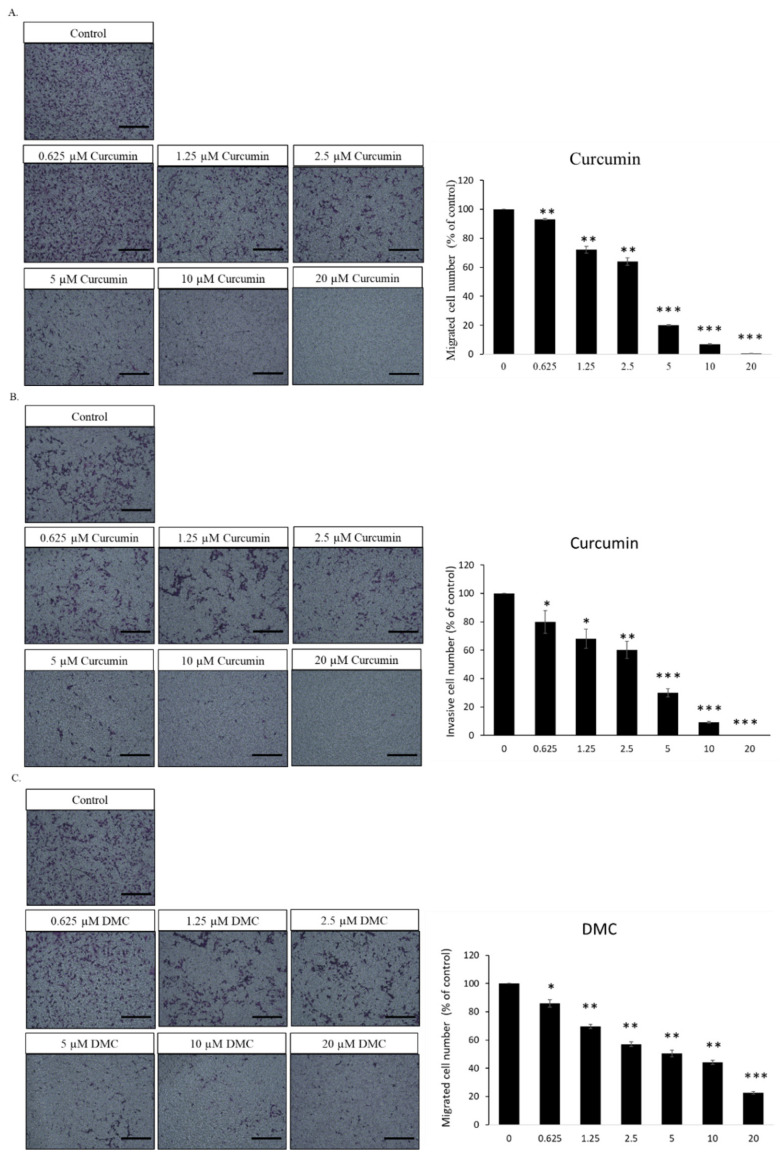
Inhibitory effects of curcuminoids (i.e., curcumin and DMC) on migration and invasion of endothelial cells. Transwell migration and Matrigel invasion assays were applied for analyzing the behavior of migration and invasion, respectively, of HUVECs administered with curcuminoids (i.e., curcumin and DMC). (**A**) The migration assay indicates that curcumin significantly inhibited the HUVECs migration. The left panel displays the image captured by a microscope at 100× magnification. The right panel displays the statistical study. (**B**) Matrigel invasion assays showed that curcumin significantly inhibited the HUVECs invasion. The left panel shows the image acquired with a microscope at 100× magnification. The right panel shows the statistical analysis. (**C**) The migration assay indicates that DMC significantly inhibited the HUVECs migration. The left panel shows the image acquired with a microscope at 100× magnification. (**D**) The Matrigel invasion assay indicates that DMC significantly inhibited the HUVECs migration. The left panel displays the image taken with a microscope at 100× magnification. * *p* < 0.05, ** *p* < 0.01, and *** *p* < 0.001 compared to the control group (treated with 10% FBS alone). All the scale bars indicate a length of 200 μM.

**Figure 5 ijms-23-03889-f005:**
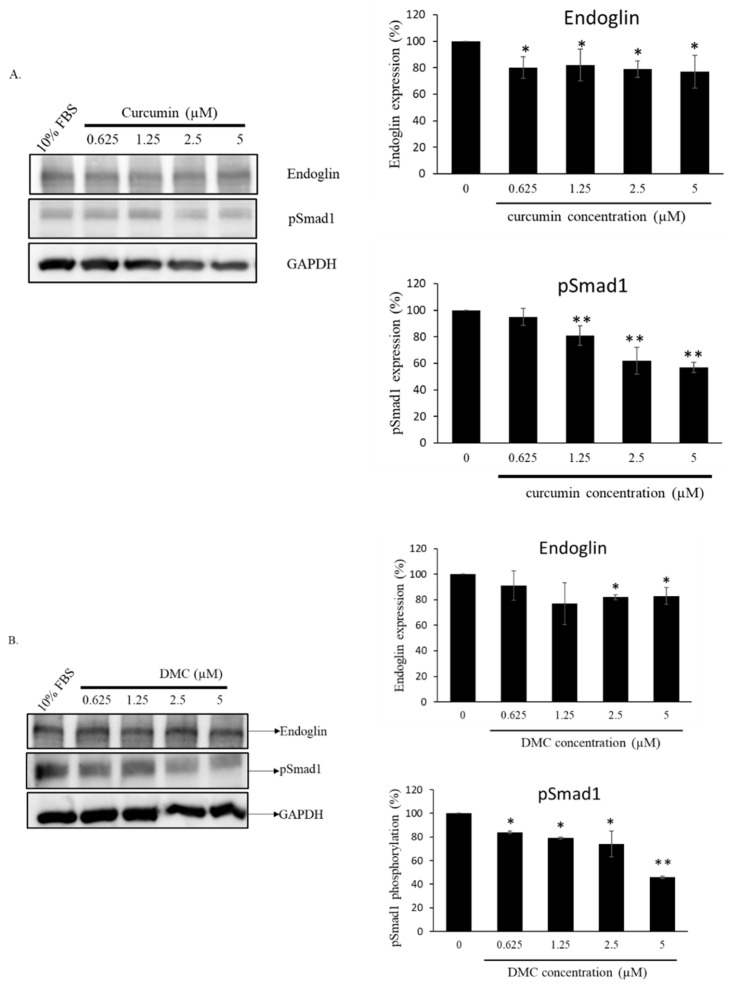
The inhibitory effects of curcuminoids (i.e., curcumin and DMC) on endoglin and phosphorylation Smad1 in HUVECs in the long term (72 h.) (**A**)Treatment with curcumin (0.625, 1.25, 2.5, and 5.0 µM) and (**B**) DMC (0.625, 1.25, 2.5, and 5.0 µM) substantially decreased endoglin expression and pSMAD1 phosphorylation. Endoglin expression and pSMAD1 phosphorylation were tested using Western blotting assays. * *p* < 0.05 and ** *p* < 0.01 compared to the control group (treated with 10% FBS alone).

**Table 1 ijms-23-03889-t001:** The screening of curcuminoids (i.e., curcumin, DMC, and bDMC) by D.S. 4.5 and values of GI_50_.

PubChem CID	Name	Structure	-CDOCKER Energy	-CDOCKER InTeraction Energy	GI_50_ (μM)
969516	Curcumin (Cur)	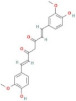	41.8465	55.3049	11.11 ± 0.08
5469424	Demethoxycurcumin (DMC)	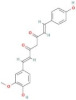	43.9046	55.2537	18.03 ± 0.46
5315472	Bisdemethoxycurcumin (bDMC)	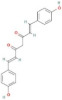	41.3177	48.5405	>100

**Table 2 ijms-23-03889-t002:** DMC and semaxanib show synergistic effects on HUVECs cell viability.

Semaxanib (25 μM)	DMC (μM)	Cell Viability (%)			
		Semaxanib (25μM)	DMC (μM)	Semaxanib + DMC	CDI
0	0	100 ± 0.08	100 ± 0.12	100 ± 1.81	
+	0.625	89.89 ± 2.50	103.58 ± 9.30	74.02 ± 2.08	0.79
+	1.25	79.78 ± 1.35	118.59 ± 5.51	65.32 ± 0.13	0.69
+	2.5	71.92 ± 1.80	119.27 ± 11.64	65.83 ± 2.94	0.77
+	5.0	69.65 ± 6.79	117.59 ± 2.31	60.64 ± 5.45	0.74
+	10.0	70.15 ± 0.33	107.56 ± 4.19	58.97 ± 3.70	0.78
+	20.0	61.70 ± 3.47	80.41 ± 7.66	40.79 ± 6.33	0.82

**Table 3 ijms-23-03889-t003:** Curcumin and semaxanib show synergistic effects on HUVECs’ cell viability.

Semaxanib (25 μM)	Cur (μM)	Cell Viability (%)			
		Semaxanib (25μM)	Cur (μM)	Semaxanib + Cur	CDI
0	0	100 ± 0.08	100 ± 0.12	100 ± 0.15	
+	0.625	89.89 ± 2.50	90.56 ± 1.79	71.60 ± 3.89	0.88
+	1.25	79.78 ± 1.35	90.71 ± 12.82	45.07 ± 0.43	0.62
+	2.5	71.92 ± 1.80	92.57 ± 2.05	47.80 ± 0.14	0.72
+	5.0	69.65 ± 6.79	89.74 ±2.29	54.44 ± 2.12	0.87
+	10.0	70.15 ± 0.33	80.98 ± 1.20	45.52 ± 0.80	0.80
+	20.0	61.70 ± 3.47	84.95 ± 9.76	39.36 ± 0.75	0.75

## Data Availability

Not applicable.

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
