# Peer review of "Curcuminoids Inhibit Angiogenic Behaviors of Human Umbilical Vein Endothelial Cells via Endoglin/Smad1 Signaling"

_ijms, 2022, doi:10.3390/ijms23073889_

Round 1

Reviewer 1 Report

The manuscript titled “Combination treatment with curcuminoids and Semaxanib inhibit human umbilical vein endothelium cells overgrowth via endoglin/Smad1 signaling” reports that the combination treatment of curcuminoids and semaxanib exhibits antiangiogenic activity in HUVECs. However, this paper is insufficient to be published in this journal because much more phenotypic analysis and molecular mechanism studies are needed. The reason is as follows.

Comments:

  1. This manuscript needs to English editing.
  2. In the Introduction, the background description of endoglin/Smad1 and VEGF is insufficient.
  3. The abstract should more faithfully represent the results of this paper.
  4. The author needs to improve the picture quality of Figure 1. The font size is small and the font quality is poor.
  5. The title of this paper is not appropriate. There is no data on overgrowth in the results to be described as overgrowth. Also, "via the inhibition of endoglin/Smad1 signaling" seems more appropriate than the title "via endoglin/Smad1 signaling".
  6. The author needs to clearly explain the purpose of the study in the introduction.
  7. To explain that cell proliferation was significantly increased by 10% FBS stimulation, the authors need to add a control group without 10% FBS treatment to the graph. “After 72 h of treatment with curcuminoids, our results demonstrated that 10% FBS stimulation significantly enhanced cell proliferation of HUVECs” (lines 109-111).
  8. The author needs to explain the results for each figure in detail and faithfully.
  9. The authors need to analyze the proliferation inhibitory activity of the 0.625~1.25 uM concentration used in the experiment.
  10. Authors need to properly position each column title in Table 2.
  11. In Figure 2, why is there no result of the combination of bDMC and semaxanib?
  12. The authors also need a cell cycle analysis on the effect of the combination of curcuminoids and semaxanib.
  13. The authors need to show the data that migration and invasion of HUVECs increased with 10% FBS compared to 0.5% FBS in the 2.4 results.
  14. The author needs to add a scale bar in Figure 4.
  15. The author needs to thoroughly discuss each result.
  16. The author needs to confirm the effect of co-treatment with curcuminoids and semaxanib in the results of 2.4
  17. The author needs to explain the relationship between Figure 5 and VEGF in the 2.5 results.
  18. How is the graph quantified in Figure 5? In Figure 5, it seems that p-Smad1 decreases similarly as the decrease of GAPDH. GAPDH densitometry should be used to normalize the values obtained for each different blotted protein. The authors need to quantify based on GAPDH.

Author Response

Comments:

  1. This manuscript needs English editing.

Answer: Thank you for your valuable suggestion. This manuscript has been carefully edited. Hopefully, it can reach the standard!!   

  1. In the Introduction, the background description of endoglin/Smad1 and VEGF is insufficient.

Answer: According to the previous study, the redundancy between endoglin and vascular endothelial growth factor (VEGF) signaling in angiogenesis and the effects of targeting both pathways on breast cancer metastasis were explored (1). Therefore, our idea is that we could find any natural-occurring compounds that could be used together with the VEGF-neutralizing agents to increase the cytotoxicity of JUVECs and reverse the drug resistance. So, the background description of endoglin/Smad1 and VEGF was carefully reviewed and described in the Introduction section as you could see in red pen. (p. 1~2; lines: 42~53).

  1. The abstract should more faithfully represent the results of this paper.

Answer: Thanks for your suggestion. The abstract has been carefully rewritten and modified as you can see in red pen. Based on our current data, the study should be focused on the inhibition of angiogenesis via the alternative angiogenic pathways (i.e. endoglin/Smad1). Therefore, we should be more focusing on the discussion about how curcuminoids, especially curcumin and demethoxycurmin (DMC) influence the endoglin/Smad1 pathway. If Cur and/or DMC could have the capacity to inhibit the endoglin/Smad1 signaling, then combination treatment with VEGF-neutralizing agents could be a future treatment plan for the cancer patients. (p. 1; written in red pen, lines 24~35).

  1. The author needs to improve the picture quality of Figure 1. The font size is small and the font quality is poor.

Answer: Thank you professor. The quality of Figure 1 has been improved. (Please see pages 4~6).

  1. The title of this paper is not appropriate. There is no data on overgrowth in the results to be described as overgrowth. Also, "via the inhibition of endoglin/Smad1 signaling" seems more appropriate than the title "via endoglin/Smad1 signaling".

Answer:    We appreciate your opinion. In this study, we should be focused on how the curcuminoids affect the angiogenic behaviors of HUVECs via the endoglin/Smad1 signaling. Because the data from the combination treatment of curcuminoids and VEGF-neutralizing agents is limited (only in Figure 2E~F). So, we thought that the title should be changed to “Curcuminoids inhibit angiogenic behaviors of human umbilical vein endothelial cells via endoglin/Smad1 signaling”. Moreover, the combination treatment of curcuminoids and semaxanib is undergoing in our laboratory. Hopefully, we could present these results soon. (Please see page 1)

  1. The author needs to clearly explain the purpose of the study in the introduction.

Answer: Thanks again for your valuable suggestion. The purpose of the study has been added to the Introduction section (shown in the last paragraph) (p. 2; lines: 84~88)

  1. To explain that cell proliferation was significantly increased by 10% FBS stimulation, the authors need to add a control group without 10% FBS treatment to the graph. “After 72 h of treatment with curcuminoids, our results demonstrated that 10% FBS stimulation significantly enhanced cell proliferation of HUVECs” (lines 109-111).

Answer: Thank you so much for your suggestion. Without the data from the 0.5% FBS treatment, cell proliferation should be used in the present study. We have corrected the expression as follows: “The cytotoxic effect of curcuminoids was examined on HUVECs treated with 10% FBS. Our results demonstrated that curcumin (Figure 2A) and DMC (Figure 2B) significantly increased the HUVECs cytotoxicity after 72 h of treatment.” I understand that we should add the 0.5% FBS control in this article. Could we answer our efforts in trying to explain how curcuminoids affect angiogenesis in the normal condition, however, your suggestion will be done in our further studies and research settings. (p. 7; lines: 127~129)

  1. The author needs to explain the results for each figure in detail and faithfully.

Answer: Yes, we have carefully rewritten and explained the results for each figure in detail and faithfully. You could see the correction in red pen for each figure.

  1. The authors need to analyze the proliferation inhibitory activity of the 0.625~1.25 μM concentration used in the experiment.

Answer: The concentration (0.625~1.25 μM) we used for the study was based on the SRB (Figure 2). From the SRB data, we found that 3.125 μM could inhibit HUVECs cytotoxicity. In view of pharmacological, we used lower concentrations (0.625~1.25 μM), which is around 0.2~0.4 times of 3.125 μM, to treat the cells. And, we found that these lower doses of curcumin and DMC could inhibit the angiogenic behaviors of HUVECs in our study. (p. 7; lines: 132~134)

  1. Authors need to properly position each column title in Table 2.

Answer: Yes, thank you so much. Each column has been properly adjusted. (Page 8). 

  1. In Figure 2, why is there no result of the combination of bDMC and semaxanib?

Answer: In this study, curcuminoids showed their inhibition of HUVECs in a dose-dependent manner, and the GI50 value of curcumin, DMC, and bDMC on HUVECs is about 11.11 μM, 18.03 μM, and > 100 μM, respectively. We found that bDMC is no effects on the cytotoxic effects on HUVECs, therefore, bDMC was excluded from the present study. (p. 7; lines: 129~131)

  1. The authors also need a cell cycle analysis on the effect of the combination of curcuminoids and semaxanib.

Answer: Per your valuable suggestion, we mentioned earlier “In this study, we were more focused on how the curcuminoids affect the angiogenic behaviors of HUVECs via endoglin/Smad1 signaling”. Therefore, in the present study, could you allow us not to present cell cycle analysis on the effect of the combination of curcuminoids and semaxanib? Due to the limited data in the combination treatment of curcuminoids and VEGF-neutralizing agents is limited (only in Figure 2E~F). So, the title should be changed to “Curcuminoids inhibit angiogenic behaviors of human umbilical vein endothelial cells via endoglin/Smad1 signaling”. However, the combination treatment of curcuminoids and semaxanib is undergoing in our laboratory.” The cell cycle analysis on the effect of the combination of curcuminoids and semaxanib is undergoing now. Hopefully, we have the chances to present our study soon.

  1. The authors need to show the data that migration and invasion of HUVECs increased with 10% FBS compared to 0.5% FBS in the 2.4 results.

Answer: We understand that we should add the 0.5% FBS control in this article. Could we answer our efforts in trying to explain how curcuminoids affect angiogenesis in the normal condition, however, your suggestion will be done in our further studies and research settings.

  1. The author needs to add a scale bar in Figure 4.

Answer: Yes, thank you so much for your great help. The scale bar has already been placed in figure 4. The scale bar indicating equals to 200 μm. (Page 11; line 197)

  1. The author needs to thoroughly discuss each result.

Answer: Yes, thank you so much for your great help. The discussion in each result has been carefully done with a red pen you could find.

  1. The author needs to confirm the effect of co-treatment with curcuminoids and semaxanib in the results of 2.4

Answer: Based on your valuable suggestion, we would like to stress the present study on how curcuminoids affect the angiogenic behaviors of HUVECs? Therefore, in the present study, could you allow us not to present the effects of combination therapy with curcuminoids and semaxanib? Combination therapy is currently undergoing. We hope the results can be revealed soon.

  1. The author needs to explain the relationship between Figure 5 and VEGF in the 2.5 results.

Answer: Thanks a lot for your suggestion. Discussion between Figure 5 and VEGF has been expressed in the discussion section. We tried to discuss their relationship as follows. Especially, our results showed that both curcumin and DMC significantly reduced the endoglin/pSmad1 protein expression (Figure 5A~B). According to the previous study, the redundancy between endoglin and VEGF signaling in angiogenesis and the effects of targeting both pathways, the endoglin-targeted inhibition will enhance the angiogenic activity of VEGF-neutralizing agents. This research strengthens our understanding that combination treatment with anti-endoglin and VEGF-neutralizing agents could enhance the anti-angiogenic effects.

  1. How is the graph quantified in Figure 5? In Figure 5, it seems that p-Smad1 decreases similarly to the decrease of GAPDH. GAPDH densitometry should be used to normalize the values obtained for each different blotted protein. The authors need to quantify based on GAPDH.

Answer: Thanks for your suggestion. So, we went to normalize the values obtained for endoglin and pSmad1 by GAPDH. And, our results demonstrated that long-term exposure (72 h) to curcumin and DMC considerably decreased the endoglin levels and phosphorylation of Smad1 in HUVECs. Especially, the endoglin/pSmad1 protein expressions were normalized by GAPTH, and, were found significantly decreased (Figure 5A~B). 

  1. M Paauwe, et al. Endoglin targeting inhibits tumor angiogenesis and metastatic spread in breast cancer, Oncogene. 2016 Aug 4;35(31):4069-79.

Reviewer 2 Report

I have read and analyzed the manuscript of Chou et al. I think that obtained results are interesting and potentially considering for publication. Nevertheless, I have some questions and critical points about the manuscript.

  1. Introduction: what about the participation of endoglin in the pathogenesis and therapy of ischemic-related diseases such as myocardial infarction, hindlimb ischemia and etc? Authors have described in Introduction some in vitro processes but did not clarify the question about physiological consequences.
  2. Figure 2, Table 2. Authors wrote that curcumin and DMC enhance the suppression of HUVECs cell viability. I can agree with this conclusion about curcumin, but the addition of DMC did not suppress cell viability (average level of suppression is equal for 25 uM semaxanib in presence both of semaxanib and DMC in the cell media). Moreover, for Figure 2 I think that authors must use multiple comparisons test. According to using of t-test, I can suggest the ANOVA using for the analysis of Figure 2 results.
  3. Figure 3. Can authors clarify how many counts are used for every graph? I think that this information should be add on the Figure 3. Why the grey and red areas are crossed in graphs? Is it the result of program approximation or smth like that? Moreover, the sums of cells in all phases are more than  100%, how do authors can explain it? Also the explanation of cell cycle analysis results is absent both in Results and in Discussion.
  4. Figure 5. Oy axis on histograms: not pSMAD1 expression but pSMAD1 phosphorylation. Also site of phosphorylation should be addressed in figure. Why authors did not measure tSMAD1 expression? It is more adequate control than GAPDH. Moreover, I think that authors should use ANOVA for statistical analysis of these data. 
  5. The critical question: the manuscript named "Combination treatment with curcuminoids and Semaxanib inhibit human umbilical vein endohelium cells overgrowth via endoglin/Smad1 signaling". However, authors used semaxinib just in one experiment (Figure 2). Other figures do not contain any mentions about semaxinib. Semaxinib should be add in all experiments (Figure 3 - Figure 5) or the title of manuscript should be improved.
  6. I think, that authors understand how the format of HUVEC subculturing is critical for angiogenesis. Why authors have used just 2D system without the checking of their proof-of-concept in the simplest 3D models?
  7. Discussion. Authors have performed in vitro study with some interesting results. Why does discussion devoted to analysis of clinical trials without discussion of possible experimental data in animal models? Why the in vitro results extrapolated on clinical trials discussion?
  8. Technical point: In raw western blot files I think that the better approach is the merge of chemiluminescence image on epi-light image. All softwares for western blots imaging allow this transformation and authors also should do it.

Author Response

Comments and Suggestions for Authors

I have read and analyzed the manuscript of Chou et al. I think that obtained results are interesting and potentially considering for publication. Nevertheless, I have some questions and critical points about the manuscript.

  1. Introduction: what about the participation of endoglin in the pathogenesis and therapy of ischemic-related diseases such as myocardial infarction, hindlimb ischemia and etc? The authors have described in Introduction some in vitro processes but did not clarify the question about physiological consequences.

Answer: Thank you so much for your valuable suggestion. Endoglin mediates the profibrotic effects of angiotensin II on cardiac fibroblasts and controls the extracellular matrix synthesis via TGF-β1 (1). As well, a study indicates that MiR-208a, responsible for the endoglin expression, is involved in cardiac hypertrophy and fibrosis (2). Because this is not correlated to our study. That is why we did not put the information in this article.  

  1. Figure 2, Table 2. The authors wrote that curcumin and DMC enhance the suppression of HUVECs cell viability. I can agree with this conclusion about curcumin, but the addition of DMC did not suppress cell viability (the average level of suppression is equal for 25 uM semaxanib in the presence both of semaxanib and DMC in the cell media). Moreover, for Figure 2 I think that authors must use multiple comparisons test. According to using of t-test, I can suggest the ANOVA using for the analysis of Figure 2 results.

Answer: Thank you so much for your valuable suggestion. According to our study, we suggested that curcumin (0.625~1.25 µM) synergistically increased the anti-angiogenic activity of semaxanib (25 µM) in HUVECs (Figure 2E). As well, DMC (0.625~1.25 µM) enhanced the anti-angiogenic activity of semaxanib (25 µM) in HUVECs (Figure 2F, Table 2B). The coefficient of drug interaction (CDI) has been calculated with the equation CDI = AB/(A × B) (AB, relative cell viability treated with the combination (i.e., curcumin/DMC and semaxanib; A or B, the relative cell viability of the groups treated with the single agents). CDI < 1 suggests a synergistic effect, CDI = 1 reveals an additive effect, and CDI > 1 reveals an antagonistic effect. (Table 2A~B). Also, our statistical analysis was evaluated by one-way ANOVA. (p. 17; lines: 403~406). It is a typo.

  1. Figure 3. Can the authors clarify how many counts are used for every graph? I think that this information should be added to Figure 3. Why the grey and red areas are crossed in graphs? Is it the result of program approximation or smth like that? Moreover, the sums of cells in all phases are more than 100%, how do authors can explain it? Also, the explanation of cell cycle analysis results is absent both in Results and in Discussion. (p. 8; lines: 152~160/p. 14: lines 294~296)

Answer: Thank you professor for your suggestion and reminder. We have added this part to the cell cycle analysis section (10,000 counts were used to determine the distribution of the cell cycle recognized by flow cytometry; page 16; lines 389~390). Most of the studies showed the areas crossed between grey and red areas as you can see in the following figure (3). The sum of the cells is not 100% is because we ran three times of the experiments, and, the mean value and standard error were reached.

We also mention the cell cycle analysis in Results and in Discussion. (p 8; lines 152~160/ p. 14; lines 294~296)

  1. Figure 5. Oy axis on histograms: not pSMAD1 expression but pSMAD1 phosphorylation. Also, the site of phosphorylation should be addressed in the figure. Why authors did not measure tSMAD1 expression? It is more adequate control than GAPDH. Moreover, I think that authors should use ANOVA for statistical analysis of these data.

Answer: Thank you so much for your valuable suggestion. tSmad1 in this article was not addressed. I realize that is very important to get the data. Thanks for your suggestion and reminder. We will absolutely add tSmad1 in our next experiment settings. Thank you so much. Also, our statistical analysis was evaluated by a one-way ANOVA. We also went to normalize the values obtained for endoglin and pSmad1 by GAPDH. And, our results demonstrated that long-term exposure (72 h) to curcumin and DMC considerably decreased the endoglin levels and phosphorylation of Smad1 in HUVECs. Especially, the endoglin/pSmad1 protein expressions were normalized by GAPTH, and, were found significantly decreased (Figure 5A~B). 

  1. The critical question: the manuscript named "Combination treatment with curcuminoids and Semaxanib inhibit human umbilical vein endothelial cells overgrowth via endoglin/Smad1 signaling". However, the authors used semaxinib just in one experiment (Figure 2). Other figures do not contain any mentions about semaxinib. Semaxinib should be added in all experiments (Figure 3 - Figure 5) or the title of the manuscript should be improved.

Answer: We appreciate your opinion. In this study, we should be focused on how the curcuminoids affect the angiogenic behaviors of HUVECs via the endoglin/Smad1 signaling. However, the data from the combination treatment of curcuminoids and VEGF-neutralizing agents is limited (only in Figure 2E~F). So, we thought that the title should be changed to “Curcuminoids inhibit angiogenic behaviors of human umbilical vein endothelial cells via endoglin/Smad1 signaling”. Moreover, the combination treatment of curcuminoids and semaxanib is undergoing in our laboratory.

  1. I think, that authors understand how the format of HUVEC subculturing is critical for angiogenesis. Why authors have used just 2D system without the checking of their proof-of-concept in the simplest 3D models?

Answer: I totally agree with your suggestion. The 3D system is basically more critical for us to do the experiments, especially the cancer research. We will absolutely take care of using 3D models as our next experiment settings.

  1. Discussion. Authors have performed in vitro study with some interesting results. Why does discussion devoted to analysis of clinical trials without discussion of possible experimental data in animal models? Why the in vitro results extrapolated on clinical trials discussion?

Answer: Thanks a lot for your suggestion my idea is that the significance of endoglin-targeting antibody is currently considered valuable. Our study focused on small molecules that may still have certain benefits. As we mentioned in the Discussion section: Although, protein drugs showed promising outcomes to antagonize the tumor angiogenesis, yet, several disadvantages have been known, including their high cost, stability, and serious immune-related adverse effects. Therefore, small molecules to modulate the endoglin/Smad1 signaling is necessary to be established and combination treatment with a VEGF antagonist to decrease drug resistance. (p. 14; lines 280~291)

  1. Technical point: In raw western blot files I think that the better approach is the merge of chemiluminescence image on the epilight image. All software for western blots imaging allow this transformation and authors also should do it.

Answer: Thanks for your suggestion. The blot imaging has been transformed. So, we went to normalize the values obtained for endoglin and pSmad1 by GAPDH. And, our results demonstrated that long-term exposure (72 h) to curcumin and DMC considerably decreased the endoglin levels and phosphorylation of Smad1 in HUVECs. Especially, the endoglin/pSmad1 protein expressions were normalized by GAPTH, and, were found significantly decreased (Figure 5A~B). 

  1. Kou-Gi Shyu, The Role of Endoglin in Myocardial Fibrosis, Acta Cardiol Sin. 2017 Sep;33(5):461-467.
  2. Kou-Gi Shyu, MicroRNA-208a Increases Myocardial Endoglin Expression and Myocardial Fibrosis in Acute Myocardial Infarction, Can J Cardiol. 2015 May;31(5):679-90.
  3. Xiaolin Bu et al., Effects of recombinant human parathyroid hormone (1-34) on cell proliferation, chemokine expression and the Hedgehog pathway in keratinocytes, 2018, Molecular Medicine Reports 17(4)

Round 2

Reviewer 1 Report

acceptable

Author Response

Comments and Suggestions for Authors

Acceptable

Dear Professor/Editor

Thank you so much for your great help. Your valuable suggestion gave us an idea to make this manuscript better. Would you please also see the following shown that this manuscript has been English edited. Thank you so much for your very valuable suggestion.

Reviewer 2 Report

I comment author's responses with saving of my initial review numbering

  1.  The absence of correlation between previously published data and your experimental data is not a reason for ignore of other authors. Authors must discuss data in light of all previous research but not only on light of “comfortable” results. References 1 and 2 for this document should be added in the manuscript and respective discussion should be add to the manuscript.
  2. Why authors used one-way ANOVA, but not two-way ANOVA? It needs explanation.

  3. Ok.
  4. Authors do not correct Oy axis legend and phosphorylation site of SMAD1 is not addressed. Using of one-way ANOVA should be clarified.

  5. Ok.
  6. Ok.
  7. Ok.
  8. Ok.

Author Response

Comments and Suggestions for Authors

I comment author's responses with saving of my initial review numbering

Dear Professor/Editor

Thank you so much for your great help. Your valuable suggestion gave us an idea to make this manuscript better. Would you please also see the following shown that this msnuscript has been English edited. Thank you so much for your very valuable suggestion.

  1.  The absence of correlation between previously published data and your experimental data is not a reason for ignore of other authors. Authors must discuss data in light of all previous research but not only on light of “comfortable” results. References 1 and 2 for this document should be added in the manuscript and respective discussion should be add to the manuscript.

Answer: Thank you so much for your helpful suggestion. We added the opposite role of endoglin on the development of angiogenesis in the Discussion section. Also, we added references 1 and 2 in the Discussion section again. So, you could find the correction at the beginning of the Discussion section in red pen (p. 13; lines: 234~247).

“The tumor vasculature is necessary for tumor growth and metastasis and becomes a major target of several anti-cancer medications [1, 28]. Therefore, targeting VEGF is used for anti-angiogenic therapy. However, VEGF-neutralizing treatment can not reach clinical outcomes, upregulation of endoglin/Smad1 signaling may play an important role. An increased endoglin expression following a VEGF-neutralizing therapy (semaxanib®) was found in vivo. Also, endoglin-targeting antibody enhanced VEGF expression in HUVECs [2]. Thus, combination treatment with an endoglin antagonist and VEGF-neutralizing agents may strengthen anti-angiogenic therapy.

Although endoglin is believed to play important role in regulating the development of angiogenesis, a study demonstrated that overexpression of endoglin does not improve angiogenesis. Nevertheless, it prevents blood vessels from maturing and stabilizing during angiogenesis. In contrary to what had been postulated, endoglin overexpression does not cause an increase of tumor vascularization, expediting the intravasation and metastasis of tumor cells [29].

  1. Why authors used one-way ANOVA, but not two-way ANOVA? It needs explanation.

Answer: Dear Professor, thank you so much. I understand that I did not clearly answer your question last time. Please allow me to explain. As I understand that “a one-way ANOVA (“analysis of variance”) compares the means of three or more independent groups to determine if there is a significant difference between the corresponding population means in one independent variable.”. In the present study, we want to realize that curcumin/DMC alone or together with semaxanib significantly affects the cell viability of HUVECs. In the figures 2 A~D: Our results showed that Cur, DMC, bDMC, and semaxanib dose-dependently affect the cell viability and presented significant difference when compared to the control. In the figure 2 E~F: Our results compared the combination treatment (Cur/DMC + semaxanib) with the semaxanib alone. And, our results showed a significant difference between the combination treatment of Cur+semaxanib and semaxanib alone. Also, our results showed a significant difference between the combination treatment of DMC+semaxanib and semaxanib alone. That is why we used a one-way ANOVA statistical analysis.

  1. Ok. Dear Professor. Thank you so much.

  1. Authors do not correct Oy axis legend and phosphorylation site of SMAD1 is not addressed. Using of one-way ANOVA should be clarified.

Answer: Thanks again for your suggestion. We are sorry that we miss this correction. However, the correction has been made as you could find from Figure 5. Also, a one-way ANOVA was conducted in this experiment. 

  1. Ok. Dear Professor. Thank you so much.
  2. Ok. Dear Professor. Thank you so much.
  3. Ok. Dear Professor. Thank you so much.
  4. Ok. Dear Professor. Thank you so much.